# Investigation of the Proportion of Diagnosed People Living with HIV/AIDS among Foreign Residents in Japan

**DOI:** 10.3390/jcm8060804

**Published:** 2019-06-05

**Authors:** Kazuki Shimizu, Hiroshi Nishiura, Akifumi Imamura

**Affiliations:** 1Graduate School of Medicine, Hokkaido University, Kita 15 Jo Nishi 7 Chome, Kita-ku, Sapporo-shi, Hokkaido 060-8638, Japan; kshimizu@eis.hokudai.ac.jp; 2CREST, Japan Science and Technology Agency, Honcho 4-1-8, Kawaguchi, Saitama 332-0012, Japan; 3Department of Infectious Diseases, Tokyo Metropolitan Cancer and Infectious Diseases Center Komagome Hospital, 3-18-22 Honkomagome, Bunkyo-ku, Tokyo 113-8677, Japan; imamura@kk.iij4u.or.jp

**Keywords:** epidemiology, foreigner, travel, migration, mobile men with money, importation, test-and-treat, laboratory diagnosis, antiretroviral therapy, treatment as prevention

## Abstract

Foreign residents represent an increasing proportion of newly diagnosed human immunodeficiency virus (HIV) infections and acquired immunodeficiency syndrome (AIDS) cases in Japan, though scant research has addressed this. This study aimed to estimate the diagnosed proportion of people living with HIV/AIDS (PLWHA) among foreign residents in Japan, covering 1990–2017 and stratifying by geographic region of the country of origin. A balance equation model was employed to statistically estimate the diagnosed proportion as a single parameter. This used published estimates of HIV incidence and prevalence, population size, visit duration, travel volume, as well as surveillance data on HIV/AIDS in Japan. The proportion varied widely by region: People from Western Europe, East Asia and the Pacific, Australia and New Zealand, and North America were underdiagnosed, while those from sub-Saharan Africa, South and South-East Asia, and Latin America were more frequently diagnosed. Overall, the diagnosed proportion of PLWHA among foreign residents in Japan has increased, but the latest estimate in 2017 was as low as 55.3%; lower than the estimate among Japanese on the order of 80% and far below the quoted goal of 90%. This finding indicates a critical need to investigate the underlying mechanisms, including disparate access to HIV testing.

## 1. Introduction

While the human immunodeficiency virus/acquired immunodeficiency syndrome (HIV/AIDS) undoubtedly remains a disease responsible for a substantial number of deaths, great progress has been made in constraining it [1]. Supported largely by widespread testing and antiretroviral therapy (ART), referred to as treatment as prevention (TasP) [2,3,4], the annual number of AIDS-related deaths worldwide decreased from 1.9 million in 2005 to less than one million in 2016. Incidence of HIV infection also decreased, from 2.8 million in 2000 to 1.8 million in 2017 [5]. Although the end of the AIDS epidemic is still far away [6], these declines show a positive association with the Joint United Nations Programme on HIV and AIDS (UNAIDS), which launched its “90-90-90” targets in 2014, aimed at ensuring that 90% of people living with HIV know their status, 90% of infected individuals receive ART, and 90% of infections under ART benefit from viral load suppression [7]. The latest evidence suggests that having an undetectable viral load on ART can ensure the absence of HIV transmission, referred to as Undetectable = Untransmittable (U = U) [8,9,10], and UNAIDS even aims to achieve “95-95-95” by 2030 [11,12]. As the risk profile of HIV infection is highly heterogeneous, a combined approach is deemed essential, including coordination of globally scaled-up prevention programs, pre-exposure prophylaxis (PrEP), and detection and treatment programs [13,14,15].

Japan has been no exception from global epidemiological trends of HIV/AIDS. While the incidence of HIV diagnoses and AIDS cases continued to steadily increase for the two decades from 1985, declines began in 2008 and 2010, respectively [16,17]. Nevertheless, published studies indicate the diagnosed proportion of HIV infections in Japan has stayed below 90% [18,19]. An analysis of first-time blood donors showed that 85.6% of infected individuals were aware of the infection at the end of 2015 [18], and an extended back-calculation study using surveillance data estimated the diagnosed proportion as of the end of 2017 ranged from 77% to 84% among Japanese nationals [19]. Even in this low-burden country with good adherence and treatment retention [18], substantial efforts are needed to achieve the 90-90-90 target [17,20]. High prevalence in men having sex with men (MSM) demonstrates a need to appropriately identify risk groups and reach out to multiple at-risk populations for testing and continued treatment [18,21,22,23].

Amid these trends, there has been notably little epidemiological research focused on the increasing proportion of foreign residents among newly diagnosed HIV infections and AIDS cases in Japan [24,25]. Cases of HIV diagnoses among foreign residents increased from 78 in 2010 to 152 in 2017, while AIDS cases increased from 33 in 2010 to 44 in 2017. The cumulative total of the two accounted for 15.8% (4526/28,614) of all diagnoses in 1990–2017. Considering that the number of foreign nationals entering Japan is increasing with time [26,27,28], Japan is certain to be exposed to an increased risk of HIV/AIDS, and in line with this, estimating the number of people living with HIV and AIDS (PLWHA) among foreign residents would play a key role in clarifying the full picture of the epidemiology of HIV/AIDS in Japan.

The present study aimed to estimate the diagnosed proportion of PLWHA among foreign residents in Japan, characterizing the estimates by the regions of the country of origin.

## 2. Materials and Methods

Japan has no individual registry of HIV-infected individuals, and the present study was premised on an analysis of surveillance data based on physicians’ notification of HIV diagnosis or AIDS cases, in accordance with the Infectious Disease Control Law, Japan. We also explored a dataset on international travel, which includes items such as visit duration and travel volume. By combining the incidence and prevalence data with the travel data, we measured the incidence and cumulative incidence of HIV infection, as person–time at risk.

### 2.1. Data Descriptions

Five different datasets from the period 1990–2017 were explored: (1) estimated incidence and prevalence of HIV infection in individual countries (inferred by the Institute of Health Metrics and Evaluation (IHME)); (2) population size in each country; (3) average length of stay in Japan, by each country of origin; (4) number of foreign nationals entering Japan, by country; and (5) yearly notification data of newly diagnosed HIV infections and AIDS cases in Japan.

Regarding the first dataset, there are generally two data sources available at a global scale. AIDSinfo, assembled by UNAIDS in 1997, is widely available [29], showing country-specific estimates. However, it has 42 countries with no available data, including the United States, Canada, and mainland China. Political controversies have prevented estimates for countries or regions closely associated with Japan, such as Taiwan and Hong Kong. As an alternative, the IHME seeks to ensure comprehensive data [30,31], offering estimates not available from AIDSinfo, though methodological differences and disagreements with UNAIDS estimates need to be clarified in the future [32]. For the second dataset, each country’s population was extracted from the United Nations Population Division [33], referring to “estimates” covering 1990–2015 and the “medium variant” in 2016–2017. The third and fourth datasets were extracted from the Annual Report of Statistics on Legal Migrants [26], from the Ministry of Justice, Japan. For the third dataset, average length of stay in Japan was calculated as the weighted average of the length. The fourth dataset referred to the entire number of entries in a given year by country. Hereafter, we define foreign residents as the total number of foreign nationals entering Japan legally, inclusive of temporary visitor, e.g., entering Japan for sightseeing, business, and visiting relatives.

Regarding the fifth dataset, as reported elsewhere [18], it is mandatory in Japan for physicians to notify the government of all newly diagnosed HIV infections and AIDS cases, as these are designated infectious diseases requiring such notice. The reporting system went into effect in 1985, and the National AIDS Surveillance Committee publicly reports the dataset [24]. We examined the yearly notification data of HIV diagnoses and AIDS cases from 1990 to 2017 involving individuals classified as being of foreign nationality. Nationalities were grouped into 10 regions: Western Europe, North Africa and Middle East, sub-Saharan Africa, South and South-East Asia, Eastern Europe and Central Asia, East Asia and Pacific, Australia and New Zealand, North America, Caribbean, and Latin America, in accordance with the classifications in the UNAIDS 1998 report [34]. Unified and separated nations over time were manually corrected to be consistent with the present-day classifications (see Appendix A).

### 2.2. Balance Equation Model

We then adopted a modeling method, described in this section. Throughout the analysis, we used incidence of HIV infection (i.e., the number of newly acquired HIV infections), *N_i_*, and PLWHA, *M_i_*, in country *i*. Our approach was premised on a balance equation following the notion that people from foreign countries possess the same risk of infection as they do in their home country. Person–time risk is calculated as the product of *D_i_* and *V_i_*, where *D_i_* represents the average length of stay in Japan and *V_i_* is the total number (volume) of foreign travelers from *i*. Supposing that the incidence in *i* is *k*_1_ times greater than that among those from *i* in Japan, the balance equation of the incidence is:(1)$$k_{1}\frac{U_{i}}{D_{i}\;V_{i}}=\frac{1}{365}\frac{N_{i}}{P_{i}}$$
where *U_i_* is the reported number of HIV diagnoses among people from *i* in Japan, and *P_i_* is the population size in *i*. Assuming the same balance applies to PLWHA, we have:(2)$$k_{2}\frac{W_{i}}{D_{i}\;V_{i}}=\frac{1}{365}\frac{M_{i}}{P_{i}}$$
where *W_i_* is the reported number of PLWHA, assumed as equal to the cumulative number of PLWHA, among people from *i* in Japan, and *k*_2_ is the relative risk of having HIV infection in the country of origin compared with that in Japan. Within each year, people in the country of origin experience a total of 365 days at risk, and each traveler from *i* experiences *D_i_* days at risk, equal to their length of stay. In Equations (1) and (2), estimates of *U_i_* and *W_i_* were extracted from surveillance data [24], while *D_i_* and *V_i_* were retrieved from statistics on legal migrants [26]. Additionally, *N_i_* and *M_i_* were obtained from IHME estimates [30], while *P_i_* was retrieved from United Nations population estimates [33].

In the surveillance data, the nationalities of newly infected HIV diagnoses were not monitored from 2011 to 2015, and only the total new HIV diagnoses and AIDS cases among foreign residents were reported during the corresponding time period. Additionally, incidence was reported by global region in other years, rather than country-specific data. As such, we revised Equations (1) and (2) to accommodate (country-)grouped data:(3)$$k_{1}^{*}\frac{U^{*}}{\sum_{i}D_{i}\;V_{i}}=\frac{1}{365s}\overset{s}{\underset{i=1}{\sum }}\frac{N_{i}}{P_{i}}$$
where *U*^*^ is the reported number of HIV diagnoses, across those from different countries, in Japan, and *s* is the total number of countries in a specified group. The same approach applies to PLWHA, i.e.:(4)$$k_{2}^{*}\frac{W^{*}}{\sum_{i}D_{i}\;V_{i}}=\frac{1}{365s}\overset{s}{\underset{i=1}{\sum }}\frac{M_{i}}{P_{i}}$$

In empirical observations, the datasets of *U* and *W* for Japan were scarce, and frequently, they were zero-valued. Thus, we assumed *U* and *W* followed zero-inflated Poisson distributions. The resulting likelihood function that allows us to estimate *k*_1_ as:(5)$$L\left(k_{1},q_{1};X=j\right)=\{\begin{array}{c} q_{1}+\left(1-q_{1}\right)exp\left(-\frac{\sum_{i}D_{i}\;V_{i}}{365sk_{1}}\overset{s}{\underset{i=1}{\sum }}\frac{N_{i}}{P_{i}}\right)\;if\;j=0\\ \left(1-q_{1}\right)\frac{exp\left(-\frac{\sum_{i}D_{i}\;V_{i}}{365sk_{1}}\sum_{i=1}^{s}\frac{N_{i}}{P_{i}}\right){\left(\frac{\sum_{i}D_{i}\;V_{i}}{365sk_{1}}\sum_{i=1}^{s}\frac{N_{i}}{P_{i}}\right)}^{j}}{j!}\;if\;j>0 \end{array}$$
where *q* is the probability of extra zeros. Similarly, we have a likelihood for estimating *k*_2_:(6)$$L\left(k_{2},q_{2};X=j\right)=\{\begin{array}{c} q_{2}+\left(1-q_{2}\right)exp\left(-\frac{\sum_{i}D_{i}\;V_{i}}{365sk_{2}}\overset{s}{\underset{i=1}{\sum }}\frac{M_{i}}{P_{i}}\right)\;if\;j=0\\ \left(1-q_{2}\right)\frac{exp\left(-\frac{\sum_{i}D_{i}\;V_{i}}{365sk_{2}}\sum_{i=1}^{s}\frac{M_{i}}{P_{i}}\right){\left(\frac{\sum_{i}D_{i}\;V_{i}}{365sk_{2}}\sum_{i=1}^{s}\frac{M_{i}}{P_{i}}\right)}^{j}}{j!}\;if\;j>0 \end{array}$$

The mean incidence from Equations (5) and (6) is $\left(1-q_{1}\right)\frac{\sum_{i}D_{i}\;V_{i}}{365sk_{1}}\overset{s}{\underset{i=1}{\sum }}\frac{N_{i}}{P_{i}}$ and $\left(1-q_{2}\right)\frac{\sum_{i}D_{i}\;V_{i}}{365sk_{2}}\overset{s}{\underset{i=1}{\sum }}\frac{M_{i}}{P_{i}}$, respectively. Likelihood functions (Equations (5) and (6)) were minimized to obtain maximum likelihood estimates.

As part of the sensitivity analysis, we also introduced an additional parameter to the balance equation to address challenges toward our key assumption, i.e., the same risk of infection among foreign residents between in Japan and in their home country:(7)$$\alpha k_{2}^{*}\frac{W^{*}}{\sum_{i}D_{i}\;V_{i}}=\frac{1}{365s}\overset{s}{\underset{i=1}{\sum }}\frac{M_{i}}{P_{i}}$$
where parameter *α* represents the ascertainment ratio. If *α* > 1, it indicates that foreign residents in Japan are more frequently diagnosed in Japan than in their home country. An alternative interpretation is that foreign residents in Japan are at greater risk of infection than the average in their home country.

### 2.3. Ethical Considerations

This study analyzed secondary datasets that were anonymized in advance and publicly available; thus, ethical approval by an institutional review board was not required.

## 3. Results

Figure 1A shows the reported number of HIV diagnoses among foreign residents in Japan; this peaked in 1992 and gradually declined by 1998, while an upward trend is seen from 2014. South and South-East Asia dominate the group, followed by Latin America and sub-Saharan Africa countries. Figure 1B shows the cumulative number of HIV diagnoses and AIDS cases among foreign residents in Japan; the total in 1990–2017 is 4526. The three most highly represented regions are South and South-East Asia (36.6%), Latin America (11.4%), and sub-Saharan Africa (6.6%). Figure 1C illustrates the estimated yearly incidence of HIV infection by region. The global HIV incidence peaked in 1999, and a marked decline set in by 2005. From 2005 to 2015, the decline leveled off, and 1.9 million people were still reported as newly acquiring HIV infection in 2017; 62.4% of new infections were reported from sub-Saharan Africa. South and South-East Asia followed, accounting for 11%, and Eastern Europe and Central Asia were third, at 9.3%. Figure 1D illustrates the temporal distribution of the yearly number of foreign nationals entering Japan by region. Except for two sudden dips—in 2009, the pandemic year and also the year following the 2008 financial crisis, and the other in 2011, when the Great East Japan Earthquake struck—there has been an upward trend since around 2000. In 2017, people from East Asia and Pacific countries constituted 72.4%, and those from South and South-East Asia ranked second (12.5%).

Figure 2 shows estimated parameter 1/*k*_2_, the diagnosed proportion of PLWHA among foreign residents in Japan over time. Two separate panels are shown because of variations in the estimates of this proportion. Figures for those from Western Europe, East Asia and Pacific, Australia and New Zealand, and North America appear to have mostly been below 100%. People from East Asia and Pacific are the most underdiagnosed group, with a high of 91.4% in 1999 and 23.1% at the latest estimate. People from North Africa and Middle East, South and South-East Asia, and sub-Saharan Africa belong to the overreported group (Figure 2B). South and South-East Asians experienced an overestimation of 991% in 1996. The proportion for Latin America began rising in 2011 and reached 829% in 2016 (Figure 2B).

Figure 3 displays the time-dependent estimate of diagnosed proportions of HIV infections and PLWHA among all foreign residents in Japan. The diagnosed proportion of new HIV infections was highest in 1992, when the Ministry of Health and Welfare, Japan revised the principles of comprehensive AIDS control strategy and launched the “Headquarter for Stop AIDS” to accelerate the Stop AIDS campaigns, estimated at 77.7% (95% confidence interval (CI): 70%, 87.3%) (Figure 3A). Apart from this, the lowest value was recorded in 2008, at 9.8% (95% CI: 8.2%, 12.3%). The latest estimate, in 2017, was 39.5% (95% CI: 34.1%, 47%). Figure 3B shows the estimates of the diagnosed proportion of PLWHA among all foreign residents. On the whole, a moderate but steady improvement is apparent over the years. The value amounted to 68.6% (95% CI: 66.6%, 70.8%) in 2014. Despite the improvements, the proportion plunged to 55.3% (95% CI: 53.8%, 57%) in 2017.

Figure 4 compares the observed and estimated HIV incidence and PLWHA among foreign residents in Japan in 2017. Estimated numbers of new HIV infections are shown by region (Figure 4A). Our calculation is restricted to six regions, because in 2017, there was no report of HIV cases for those from the Western Europe, North Africa and Middle East, Eastern Europe and Central Asia, and Caribbean regions. A remarkable difference between observed and estimated numbers is seen for the East Asia and Pacific regions. While the reported number of cases was 27 (green bar), the estimated number was 96.8 (95% CI: 60.3, 133.3). Those from South and South-East Asia, sub-Saharan Africa, and North America showed similar trends: 29, six, and three infections were reported for these regions, respectively, and estimated HIV infections were 32 (95% CI: 20.4, 43.7), 5.0 (95% CI: 1, 9), and 6.8 (95% CI: 0, 14.6). Figure 4B compares the estimated PLWHA against observed cumulative incidence. Globally, estimated PLWHA among foreign residents in 1990–2017 was 5099.6 (95% CI: 4951, 5248.1), while 2822 cases were observed, exclusive of “unknown” nationals. A similar difference appeared in Western Europe and East Asia and Pacific; PLWHA amounted to 49 and 181, respectively, whereas the estimated numbers were 98.8 (95% CI: 71.2, 126.5) and 782.8 (95% CI: 668.7, 896.8). The observed data were higher than the upper 95% confidence intervals in sub-Saharan Africa, South and South-East Asia, and Latin America, indicating people from these regions have been more effectively diagnosed than those from other regions.

As we see in Figure 2, sometimes, the estimated diagnosed proportion of PLWHA exceeded 100%, and thus, we examined the sensitivity of our estimated diagnosed proportion to under-ascertainment. Figure 5 graphically shows how the latest estimate of the diagnosed proportion, 55.3%, is sensitive to the variation in *α*, the ascertainment ratio. As anticipated from Equation (7), the resulting estimated diagnosed proportion is linearly dependent on *α*. Estimates based on additional methods (e.g., using CD4 count data) should validate ours to address this uncertainty.

## 4. Discussion

In the present study, we estimated the diagnosed proportion of HIV incidence and PLWHA among foreign residents in Japan in 1990–2017. We used a simplistic balance equation modeling approach, assuming the risk of HIV infection among foreign residents was identical to the calculated population average of the risk in their country of origin. The balance relationship was exploited to derive the key estimate of the diagnosed proportion as a parameter via maximum likelihood estimation. This exercise revealed the diagnosed proportion widely varied by geographic regions of the country of origin. People from Western Europe, East Asia and Pacific, Australia and New Zealand, and North America were underdiagnosed. The overall diagnosed proportion of PLWHA among foreign residents in Japan evidently improved over time, but the latest estimate in 2017 was as low as 55.3%, which is well below the UNAIDS 90% target [7].

To our knowledge, this study is the first to provide statistical estimates of the diagnosed proportion of HIV infections and PLWHA among foreign residents in Japan. Three important conclusions can be drawn. First, among these foreign residents, the first 90 of the UNAIDS 90-90-90 goal was far from satisfied. Given that published estimates of the diagnosed proportion among the Japanese have been around 80% in 2017 [19], the estimate among foreign residents is far lower, indicating a critical need to investigate the underlying mechanisms. Second, the estimated diagnosed proportion as of 2017 was 55.3%, though on the whole and over time, it has improved (Figure 3B), far exceeding 50% in the recently reported 6 years. Third, the diagnosed proportion of PLWHA among foreign residents greatly differed by geographic region of their country of origin. The estimated number of PLWHA clearly showed HIV infection among those from South and South-East Asia was more likely to be diagnosed than in other groups. HIV among residents from sub-Saharan Africa and Latin America was also more effectively diagnosed than for those from other regions. HIV in residents from higher-income and low-HIV-prevalence countries, which contain the majority of industrialized countries (Western Europe, Australia and New Zealand, and North America), tended to be underdiagnosed. Of course, varying estimates reflect not only different rates of diagnosis but also heterogeneous characteristics of residents—e.g., mobile men with money are recognized as one of the at-risk groups [35,36]—and the movement of high-risk individuals across areas with completely different HIV prevalence can drive HIV transmission [37]. Employing a range of tailored interventions depending on residents’ risk profiles, including task shifting to specific foreign subpopulations based on their social, cultural, and traditional norms, could lead to successful results [38,39,40,41], and our findings indicate that detailed approaches would have to be considered.

Amid wide-scale globalization, the number of foreign residents in Japan continues to rise [27]. Adapting to them has become a political challenge, because their disease backgrounds and risk factors widely differ from those in Japan, not to mention that foreign residents are likely to experience barriers to accessing health services [42]. The issue of HIV patients among foreign residents is widely recognized in Western Europe and the United States. Higher incidence of HIV diagnoses and late presentation are major characteristics of foreign residents who live in countries with a high income and low HIV prevalence [43,44]. In general, foreign residents from high-prevalence countries, including those from sub-Saharan Africa, present a higher rate of HIV diagnoses [45,46,47]. However, new HIV diagnoses among foreign residents from low-prevalence countries represent an increasing proportion in Europe [43,48]. A recent study from Australia mentioned that infections with advanced clinical symptoms were mostly observed among migrants from South-East Asia, and this has placed a great deal of strain on the government [38,49]. Although the estimated magnitude is not yet substantial, people from East Asia are considered the imminent majority of residents for the next decades because of the region’s sizeable population. HIV incidence has been estimated as decreasing in East Asian countries, but it should be remembered that the absolute number of new HIV infections in China, Japan’s neighbor, exceeded 30,000 in 2017 [30]. Furthermore, it has become clear that sex tourism poses a substantial risk of infection. Sex tourism leads to risky sexual behaviors, such as unprotected anal intercourse, forgoing condom use, and other behaviors that increase HIV risk [50,51,52], amplifying regional HIV epidemics [53]. Considerable change in sexual behaviors has been observed through modernization in Asian cities such as Hanoi, Shanghai, and Taipei [54].

Leaving patients undiagnosed and untreated serves as a risk for secondary transmission of HIV [25]. TasP and U = U have been widely acknowledged, and early initiation of ART and successful treatment of HIV could reduce or even eliminate sexual transmission, yielding both direct and indirect benefits [2,3,4,8,9,10,55]. Despite various barriers, voluntary counselling and HIV testing are required for foreign residents to receive early diagnosis, overcome HIV-related stigma, and bring preferable clinical outcomes [56,57,58,59]. At the same time, while opt-out screening originally acted as a useful tactic for eliminating mother-to-child transmission in high-prevalence settings [60], its cost-effectiveness in low-prevalence settings has also been reported [61]. The United States practically adopted an opt-out approach in 2006 [62]. The benefit has also been identified in the United Kingdom [63]. The introduction of provider-initiated opt-out HIV screening is, however, a debatable topic.

Five limitations should be noted in the present study. First, it was premised on a balance equation, ignoring human mobility and using only average values by nationality. Especially for the latter, the risk profile of HIV is known to be heterogeneous (e.g., in Japan, geographic heterogeneity in risk of infection is expected). Second, being diagnosed with HIV is a potential cause for foreigners to leave Japan; thus, the estimated PLWHA may considerably differ from the actual number. Notably, our PLWHA calculation was almost equivalent to the cumulative incidence of HIV infection, ignoring natural mortality. We simply excluded these points, as we did not have sufficient empirical datasets to explicitly quantify the processes of foreign residents’ natural and social migration. Third, we had several data limitations, relying solely on epidemiological surveillance data spanning 1990–2017; this did not show country-specific data but only data that classified country of origin into 10 regions. Moreover, the present surveillance did not consistently pursue nationality data, especially for 2011–2015, and there was a substantial number of “unknown” nationalities in the empirical data. Although collection restarted from 2016, 82.8% (140/169) in 2016 and 42.9% (84/196) in 2017 of HIV diagnoses among foreign residents were still reported as “unknown” nationality. Given our findings indicating a critical need to more deeply examine foreign residents and considering the heterogeneity of transmission [64,65,66], scrutinizing nationality data is vital [38,39,40], potentially constituting an important component for establishing an effective country-specific tailored approach to PLWHA. Fourth, the balance equation implicitly assumed the risk of infection was determined by the risk in the country of origin among residents, ignoring possible new infections in Japan. This may contradict certain documented evidence [56,67,68,69]. Fifth, we omitted various critical pieces of information, including patients’ age at diagnosis, sex, mode of exposure (e.g., heterosexual, homosexual, intravenous drug user, mother-to-child), year of arrival, visa status, and duration of residence, which are essential for constructing evidence-based (and possibly tailored) testing, treatment, and care programs. More comprehensive details of risk profiles could have been identified if the National HIV Registry recorded these characteristics.

Designing effective programs that maximize access to HIV testing, treatment, and care to reduce HIV transmission, as well as early engagement of political leadership, are imperative [70]. While many obstacles exist for more meticulous approaches, the present study provided essential perspectives on HIV diagnoses among foreign residents in Japan that were interpretable from existing empirical data. In a globalized world with large-scale human mobility, our estimates of the diagnosed proportion of HIV infections and PLWHA among foreign residents can serve as impetus for considering better access for testing and care in the future in Japan.

## 5. Conclusions

The present study estimated the diagnosed proportion of HIV incidence and PLWHA among foreign residents in Japan in 1990–2017. A simplistic balance equation modeling approach was employed, revealing the diagnosed proportion widely varied by geographic region of the country of origin. People from Western Europe, East Asia and Pacific, Australia and New Zealand, and North America were underdiagnosed. The overall diagnosed proportion of PLWHA among foreign residents in Japan evidently improved over time, but the latest estimate, in 2017, was as low as 55.3%, which is well below the UNAIDS target of 90%.

## Figures and Tables

**Figure 1 jcm-08-00804-f001:**
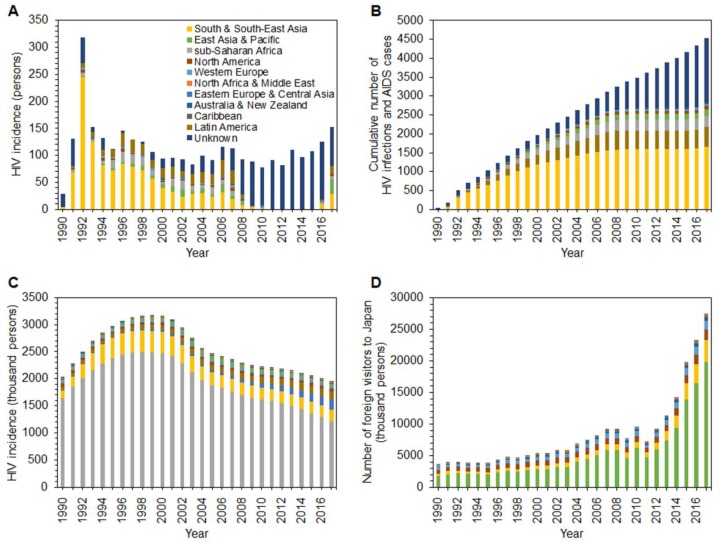
Human immunodeficiency virus (HIV) and nationality in Japan and worldwide, 1990–2017. (**A**) Yearly incidence (i.e., the number of new infections) of HIV among foreign residents in Japan. (**B**) Cumulative number of HIV infections and acquired immunodeficiency syndrome (AIDS) cases among foreign residents in Japan. (**C**). Estimated yearly incidence of HIV infections in the home country, by region (as estimated by the Institute of Health Metrics and Evaluation). (**D**) Temporal distribution of the yearly number of foreign visitors to Japan, by region of the country of origin.

**Figure 2 jcm-08-00804-f002:**
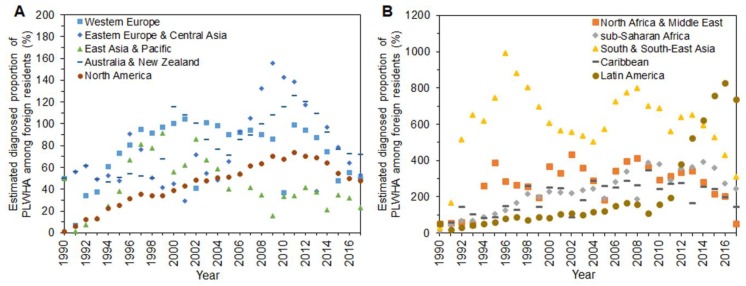
Estimated diagnosed proportion of human immunodeficiency virus (HIV) infections among foreign residents in Japan, 1990–2017. Estimates of the proportion of diagnosed infections out of the cumulative number of HIV infections and acquired immunodeficiency syndrome (AIDS) cases. Two panels are shown because of varying estimates. (**A**) People from Western Europe, Eastern Europe and Central Asia, East Asia and Pacific, Australia and New Zealand, and North America. (**B**) People from North Africa and Middle East, sub-Saharan Africa, South and South-East Asia, Caribbean, and Latin America.

**Figure 3 jcm-08-00804-f003:**
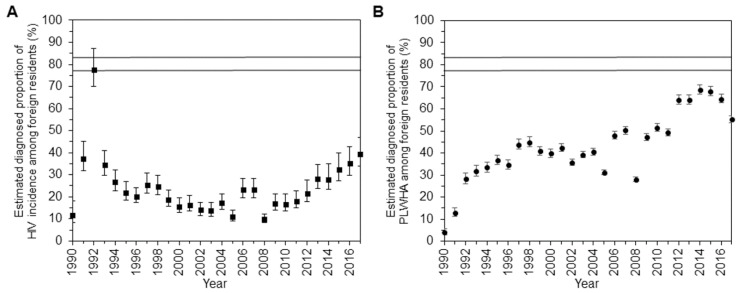
Overall estimate of diagnosed proportion of human immunodeficiency virus (HIV) infections among foreign residents in Japan, 1990–2017. (**A**) Estimates of the diagnosed proportion of the yearly incidence of HIV infections. (**B**) Estimates of the diagnosed proportion of the cumulative number of HIV infections and acquired immunodeficiency syndrome (AIDS) cases in 1990–2017. In both (**A**,**B**), whiskers extend to the upper and lower 95% confidence intervals. Two horizontal lines show the estimated diagnosed proportions among Japanese national; 77–84% [19].

**Figure 4 jcm-08-00804-f004:**
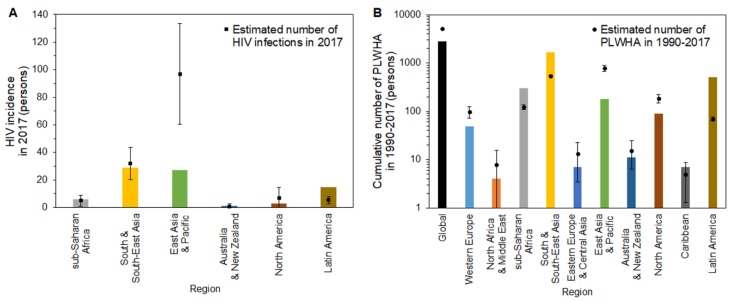
Comparison of observed and estimated HIV incidence and people living with human immunodeficiency virus (HIV)/acquired immunodeficiency syndrome (AIDS; PLWHA) among foreign residents in Japan. (**A**) Bars represent the observed incidence of HIV diagnoses in Japan in 2017, by region of the country of origin. Solid squares represent the estimated number of HIV diagnoses when using our model. (**B**) Bars represent the observed cumulative number of HIV infections and AIDS cases in Japan from 1990–2017, by region of the country of origin. Solid circles show the estimated cumulative number of HIV infections and AIDS cases by region in Japan when using our model. Whiskers extend to the upper and lower 95% confidence intervals. It should be noted that a common logarithmic scale is used on the vertical axis.

**Figure 5 jcm-08-00804-f005:**
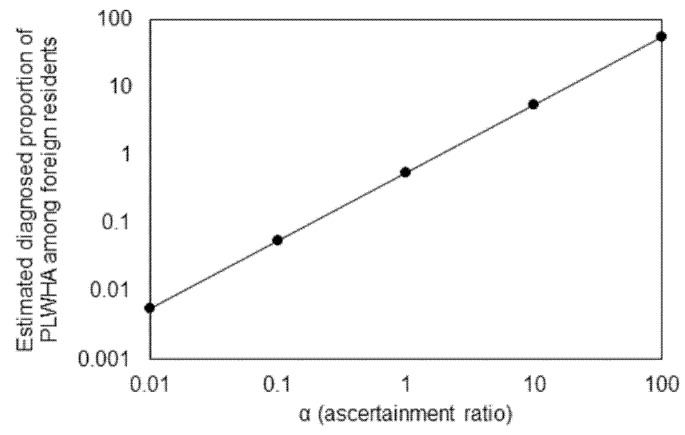
Sensitivity of estimate of diagnosed proportion of the cumulative number of HIV infections and acquired immunodeficiency syndrome (AIDS) cases in 2017 to variations in ascertainment ratio. It should be noted that a common logarithmic scale is used on both horizontal and vertical axes. Further, it should be noted that the vertical axis is the fraction (i.e., the value 1 is equivalent to 100%).

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
