# Peer review of "Investigation of the Proportion of Diagnosed People Living with HIV/AIDS among Foreign Residents in Japan"

_jcm, 2019, doi:10.3390/jcm8060804_

Round 1
Reviewer 1 Report
In their present manuscript, Shimizu and colleagues employed a balance equation model to estimate the percentage of diagnosed individuals living with HIV/AIDS among foreign residents in Japan. Their estimates suggest that about 55.3% of all HIV-infected foreigners in Japan know their status (2017). The percentage has been increasing in recent years and is lowest in people from low-HIV-prevalence countries (Western Europe, Australia and New Zealand, North America).
Major points:
1) While the authors have made great efforts to optimize the quality of their estimations, there are a few notable limitations, most of which are also mentioned in lines 314-333 of the manuscript. While they acknowledge that “being diagnosed with HIV is a potential cause for foreigners to leave Japan” (lines 317-318), diagnosed individuals may also enter Japan more or less frequently than uninfected or undiagnosed individuals.
As both, risk of HIV infection and international mobility most likely depend on the socio-economic status of an individual, it seems rather unlikely that foreign residents in Japan “possess the same risk of infection as they do in their home countries (line 111).”
Particularly Fig. 2B raises concerns regarding the quality of the estimates as the estimated proportion of diagnosed people living with HIV/AIDS (PLWHA) is above 100 % in most of the cases, reaching 1000% in individuals from South and South East Asia in 1996. Similarly, the estimated number of PLWHA is sometimes considerably lower than the actual numbers (Fig. 4B).
2) In some cases, it would be nice to include data or estimates for the Japanese population for direct comparison:
- Fig. 3A, B: Can the authors add the estimated diagnosed proportion of the Japanese population for comparison?
- Lines 23-24: “the diagnosed proportion of PLWHA among foreign residents in Japan has increased, but the latest estimate in 2017 was as low as 55.3%”. The diagnosed proportion among Japanese (in 2017) may be mentioned for comparison.
Minor points:
1) How are “foreign residents” (lines 3, 13, ..) and “foreign visitors” (e.g. line 163) defined? Does this include every foreigner in Japan, including tourists?
2) In the x-axes of Figs. 2 and 3, the authors should clearly indicate that the “estimated diagnosed proportion” is shown.
3) Lines 28-29: „HIV“, „AIDS“ may be included in the list of keywords.
4) Lines 60-65: “Considering that the number of international visitors to Japan has been dramatically increasing [22], and given that population decline in Japan has impelled the government to accept far more incoming foreign residents to bolster the labor market [23,24], the country is certain to be exposed to an increased risk of HIV/AIDS via international migration. HIV-related restrictions on entry, stay, and residence were seen in at least 27 countries as of early 2019 [25–27].” Consider to delete this section as it may be misused for political reasons.
5) Line 143-144, Fig. 1A: “South and South-East Asia dominate the group, followed by East Asia and Pacific countries” This is only true for 2017. In earlier years, South and South-East Asia were followed by Latin America and sub-Saharan Africa. This should be clarified.
6) Line 147: “Latin America (36.6%)”. Is this number correct? The proportion appears to be much lower in Fig. 1B.
7) Lines 184-185, Fig. 3A: Do the authors have any explanation for the strong increase in 1992?
8) Lines 239-240: “Given that published estimates of the diagnosed proportion among Japanese have been around 80%”. Is this an estimate for the year 2017?
9) The Appendix should be carefully rechecked for correctness, e.g.:
a. line 393: “North Macedonia” instead of “The former Yugoslav Republic of Macedonia”
b. line 398: “Botswana” instead of “Botana”
Author Response
Point-by-point responses to the reviewers (“Investigation of the proportion of diagnosed people living with HIV/AIDS among foreign residents in Japan”; Manuscript ID: jcm-513271)
[Response to the Reviewer 1]
In their present manuscript, Shimizu and colleagues employed a balance equation model to estimate the percentage of diagnosed individuals living with HIV/AIDS among foreign residents in Japan. Their estimates suggest that about 55.3% of all HIV-infected foreigners in Japan know their status (2017). The percentage has been increasing in recent years and is lowest in people from low-HIV-prevalence countries (Western Europe, Australia and New Zealand, North America).
Major points:
1) While the authors have made great efforts to optimize the quality of their estimations, there are a few notable limitations, most of which are also mentioned in lines 314-333 of the manuscript. While they acknowledge that “being diagnosed with HIV is a potential cause for foreigners to leave Japan” (lines 317-318), diagnosed individuals may also enter Japan more or less frequently than uninfected or undiagnosed individuals. As both, risk of HIV infection and international mobility most likely depend on the socio-economic status of an individual, it seems rather unlikely that foreign residents in Japan “possess the same risk of infection as they do in their home countries (line 111).” Particularly Fig. 2B raises concerns regarding the quality of the estimates as the estimated proportion of diagnosed people living with HIV/AIDS (PLWHA) is above 100 % in most of the cases, reaching 1000% in individuals from South and South East Asia in 1996. Similarly, the estimated number of PLWHA is sometimes considerably lower than the actual numbers (Fig. 4B).
We agree with the reviewer. The identical risk by nationality has been a very heavy assumption in our study. Identifying heterogeneous structure of infection among foreign residents would be of utmost importance. Unfortunately, we do not have substantial datasets to directly address this matter in the present study. However, we have been able to conduct a sensitivity analysis of the estimated diagnosed proportion to the variation in ascertainment ratio. We have added an analysis to Results (Figure 5) with hones explanations on this matter from Page 4, Lines 139-145, and Page 7, Line 237–Page 8, Line 247, clarifying that our estimate would be linearly sensitive to the ascertainment rate and thus should be validated in the future using additional datasets.
2) In some cases, it would be nice to include data or estimates for the Japanese population for direct comparison:
-Fig. 3A, B: Can the authors add the estimated diagnosed proportion of the Japanese population for comparison?
-Lines 23-24: “the diagnosed proportion of PLWHA among foreign residents in Japan has increased, but the latest estimate in 2017 was as low as 55.3%”. The diagnosed proportion among Japanese (in 2017) may be mentioned for comparison.
We thank the reviewer for these suggestions. Reference values for Japanese national were added to Figure 3, which we believe have been greatly improved in terms of the readability (Page 6, Lines 202-208). Moreover, we mentioned that about 80% of Japanese people living with HIV are diagnosed in Abstract (Page 1, Lines 24-25).
Minor points:
1) How are “foreign residents” (lines 3, 13, ..) and “foreign visitors” (e.g. line 163) defined? Does this include every foreigner in Japan, including tourists?
We thank the reviewer for noting this point. We added our definition of foreign residents as the total number of foreign nationals entering Japan legally, inclusive of “Temporary Visitor” (i.e. for sightseeing, business and visiting relatives) in Page 3, Lines 94-96. Besides, we also had to use foreign visitors that represent the total number of entries in Figure 1 (Page 5).
2) In the x-axes of Figs. 2 and 3, the authors should clearly indicate that the “estimated diagnosed proportion” is shown.
We thank the reviewer for this comment. We have revised the vertical axes of Figures 2 and 3 (Page 6).
3) Lines 28-29: „HIV“, „AIDS“ may be included in the list of keywords.
We have checked the use of keywords for this journal, and it appears that any words that are used either in the Title or Abstract can be detected by search engine. Thus, we decided to limit ourselves to include keywords that are unused in Title and Abstract.
4) Lines 60-65: “Considering that the number of international visitors to Japan has been dramatically increasing [22], and given that population decline in Japan has impelled the government to accept far more incoming foreign residents to bolster the labor market [23,24], the country is certain to be exposed to an increased risk of HIV/AIDS via international migration. HIV-related restrictions on entry, stay, and residence were seen in at least 27 countries as of early 2019 [25–27].” Consider to delete this section as it may be misused for political reasons.
We have rewritten the corresponding sentence (Page 2, Lines 62-64).
5) Line 143-144, Fig. 1A: “South and South-East Asia dominate the group, followed by East Asia and Pacific countries” This is only true for 2017. In earlier years, South and South-East Asia were followed by Latin America and sub-Saharan Africa. This should be clarified.
We agree and corrected the manuscript accordingly (Page 4, Line 152).
6) Line 147: “Latin America (36.6%)”. Is this number correct? The proportion appears to be much lower in Fig. 1B.
We apologize for the confusion and corrected the quoted numbers (Page 4, Line 155).
7) Lines 184-185, Fig. 3A: Do the authors have any explanation for the strong increase in 1992?
As a possible explanation, we have mentioned that 1992 was when the Ministry of Health and Welfare, Japan revised the principles of comprehensive AIDS control strategy and launched the “Headquarter for Stop AIDS” to accelerate the Stop AIDS campaigns (Page 6, Lines 194-196).
8) Lines 239-240: “Given that published estimates of the diagnosed proportion among Japanese have been around 80%”. Is this an estimate for the year 2017?
We have noted that the latest estimate was 2017 (Page 8, Line 263).
9) The Appendix should be carefully rechecked for correctness, e.g.: a. line 393: “North Macedonia” instead of “The former Yugoslav Republic of Macedonia”
b. line 398: “Botswana” instead of “Botana”
We thank the reviewer for carefully checking those and the suggested points were corrected accordingly (Page 11, Lines 396 and 400).
Reviewer 2 Report
In the reviewed manuscript, Kazuki Shimizuet colleagues report a modelling study estimating the proportion of foreign residents diagnosed and living with HIV in Japan. The authors estimated the proportion of HIV incidence and PLWHA among foreigner vs native residents in Japan over a 17 years period (i.e 1990-2017), using a specific modelling approach and assuming the risk of HIV infection among foreign residents was biased by the relative risk of infection in their country of origin. The study adds significant info re the numbers and future estimates about HIV incidence and PLWA in Japan, which are important especially due to the global efforts to reach WHO/UNAIDS 95-95-95 target by 2030. The presented work only answers re the 1st 95 and there is no info re the treatment proportions and/or proportions of suppressive ART among foreign vs native residents in Japan, which is also important in the U=U era. Overall, the ms is well designed and presented and to my personal opinion deserves to be published at the Journal of Clinical Medicine after minor revision.
Minor comments/suggestions
- Introduction segment; the authors should add info re the U=U statement (“the 3rd 95” goal), as in the U=U era, we know that viral suppression due to effective ART constitutes the most effective strategy for preventing onward transmission of infection and, thus, expansion of the HIV epidemic [e.g PARTNER, Opposites Attract, HPTN052 studies etc]
- Discussion segment; to my personal opinion is quite extensive and not easy to follow; I would suggest to cut some repeated info and make it more concise
Author Response
Point-by-point responses to the reviewers (“Investigation of the proportion of diagnosed people living with HIV/AIDS among foreign residents in Japan”; Manuscript ID: jcm-513271)
[Response to the Reviewer 2]
Comments and Suggestions for Authors. In the reviewed manuscript, Kazuki Shimizu et colleagues report a modelling study estimating the proportion of foreign residents diagnosed and living with HIV in Japan. The authors estimated the proportion of HIV incidence and PLWHA among foreigner vs native residents in Japan over a 17 years period (i.e 1990-2017), using a specific modelling approach and assuming the risk of HIV infection among foreign residents was biased by the relative risk of infection in their country of origin. The study adds significant info re the numbers and future estimates about HIV incidence and PLWA in Japan, which are important especially due to the global efforts to reach WHO/UNAIDS 95-95-95 target by 2030. The presented work only answers re the 1st 95 and there is no info re the treatment proportions and/or proportions of suppressive ART among foreign vs native residents in Japan, which is also important in the U=U era. Overall, the ms is well designed and presented and to my personal opinion deserves to be published at the Journal of Clinical Medicine after minor revision.
Minor comments/suggestions: -Introduction segment; the authors should add info re the U=U statement (“the 3rd 95” goal), as in the U=U era, we know that viral suppression due to effective ART constitutes the most effective strategy for preventing onward transmission of infection and, thus, expansion of the HIV epidemic [e.g PARTNER, Opposites Attract, HPTN052 studies etc]
We thank the reviewer for this positive comment. We have mentioned “U=U” in Introduction (Page 1, Lines 41-43) and Discussion (Page 9, Lines 301-303).
-Discussion segment; to my personal opinion is quite extensive and not easy to follow; I would suggest to cut some repeated info and make it more concise.
We agree with the reviewer and reduced the total volume of Discussion (e.g. Page 8, Lines 276-279 and Page 9, Lines 289, 291-5, 303-305, 320-326, 335-336).
Round 2
Reviewer 1 Report
The authors have addressed all of my initial concerns. Besides a few textual changes and corrections, they have included a definition of "foreign residents" and reference values for Japanese nationals. Furthermore, they have added a sensitivity analyses of the diagnosed proportion to the variation in ascertainment ratio. While this analysis (new Fig. 5) does not improve the (potentially poor) estimates in the present manuscript, I feel that the authors made the best of the data sets that were available.